# Effects of Coumarinyl Schiff Bases against Phytopathogenic Fungi, the Soil-Beneficial Bacteria and Entomopathogenic Nematodes: Deeper Insight into the Mechanism of Action

**DOI:** 10.3390/molecules27072196

**Published:** 2022-03-28

**Authors:** Vesna Rastija, Karolina Vrandečić, Jasenka Ćosić, Gabriella Kanižai Šarić, Ivana Majić, Dejan Agić, Domagoj Šubarić, Maja Karnaš, Drago Bešlo, Mario Komar, Maja Molnar

**Affiliations:** 1Faculty of Agrobiotechnical Sciences Osijek, Josip Juraj Strossmayer University of Osijek, Vladimira Preloga 1, 31000 Osijek, Croatia; vesna.rastija@fazos.hr (V.R.); karolina.vrandecic@fazos.hr (K.V.); jasenka.cosic@fazos.hr (J.Ć.); gabriella.kanizai@fazos.hr (G.K.Š.); ivana.majic@fazos.hr (I.M.); domagoj.subaric@fazos.hr (D.Š.); mkarnas@fazos.hr (M.K.); drago.beslo@fazos.hr (D.B.); 2Faculty of Food Technology Osijek, Josip Juraj Strossmayer University of Osijek, Franje Kuhaca 20, 31000 Osijek, Croatia; mario.komar@ptfos.hr (M.K.); maja.molnar@ptfos.hr (M.M.)

**Keywords:** coumarins, plant protection, biological activity, molecular docking, toxicity, acetylcholinesterase

## Abstract

Coumarin derivatives have been reported as strong antifungal agents against various phytopathogenic fungi. In this study, inhibitory effects of nine coumarinyl Schiff bases were evaluated against the plant pathogenic fungi (*Fusarium oxysporum* f. sp. *lycopersici*, *Fusarium culmorum*, *Macrophomina phaseolina* and *Sclerotinia sclerotiourum*). The compounds were demonstrated to be efficient antifungal agents against *Macrophomina phaseolina*. The results of molecular docking on the six enzymes related to the antifungal activity suggested that the tested compounds act against plant pathogenic fungi, inhibiting plant cell-wall-degrading enzymes such as endoglucanase I and pectinase. Neither compound exhibited inhibitory effects against two beneficial bacteria (*Bacillus mycoides* and *Bradyrhizobium japonicum*) and two entomopathogenic nematodes. However, compound **9** was lethal (46.25%) for nematode *Heterorhabditis bacteriophora* and showed an inhibitory effect against acetylcholinesterase (AChE) (31.45%), confirming the relationship between these two activities. Calculated toxicity and the pesticide-likeness study showed that compound **9** was the least lipophilic compound with the highest aquatic toxicity. A molecular docking study showed that compounds **9** and **8** bind directly to the active site of AChE. Coumarinyl Schiff bases are promising active components of plant protection products, safe for the environment, human health, and nontarget organisms.

## 1. Introduction

Fungal plant pathogens are dominant causal agents of plant diseases. Although in modern agriculture most fungal pathogens are controlled by modern crop management, fungal epidemics still occur, resulting in huge yield losses that have significant negative effects on the economy and society [1]. Nine *Fusarium* species and *Microdohium nivale* were identified from isolates of wheat plants in Croatia [2], while in eastern Croatia *F. graminearum* and *F. culmorum* are dominant (Wm. G. Sm.) [3].

Although the current use of pesticides in agriculture has led to a significant improvement in crop yields, future crop management will require the development of novel active agrochemicals that have ultra-high efficiency and are environmentally safe. Therefore, research on the multiple biological effects of newly synthesized compounds, including a test for the environmental properties, toxicity, and broad-spectrum mode of action, is necessary [4,5].

The development of new pesticides from the synthesis of active compounds to their commercialization is a costly and time-consuming process. Computer-aided molecular design (CAMD) was extensively applied in the ecotoxicological modeling and design of agrochemicals for its high efficiency in the design of new compounds, saving both time and economic costs in the large-scale experimental synthesis and biological tests. Thus, quantitative structure–activity relationships (QSAR) models that are used to predict the activity of agrochemicals serve as a reliable tool for searching for molecules with improved activity. Molecular docking and molecular dynamics simulations were explored to identify the mode of interaction between the active molecule and the target enzyme or protein related to the observed biological activity. In addition, QSAR has the potential to estimate the risks of chemicals for the environment and human health, reducing the time, monetary cost, and necessary animal testing [6].

Coumarins, secondary plant products and their synthetic derivatives, demonstrated a wide range of biological activities on different organisms (invertebrate pests, pathogenic fungus and other microorganisms and weeds). Several coumarin derivatives have been reported as strong antifungal agents against *Sclerotinia sclerotiorum*, *Botrytis cinerea*, *Colletotrichum gloeosporioides*, *Fusarium oxysporum*, *Valsa mali* and *Moniliophthora perniciosa* [7,8,9]. Coumarins also have antibacterial potential against phytopathogens: *Ralstonia solanacearum* [10], *Agrobacterium tumefaciens* [11], and *Pseudomonas aeruginosa* [12]. In addition, coumarins have nematicidal properties against the most destructive plant parasitic nematodes from genus *Meloidogyne* as well as *Bursaphelenshus* spp. and *Ditylenchus* sp. [13,14]. Moreover, Souza et al. [15] found greater or lesser potency of coumarin derivatives on bacterial Gram-positive and Gram-negative representatives *Staphylococcus aureus*, *Bacillus cereus*, *Escherichia coli*, and *Pseudomonas aeruginosa*.

Anthelmintic activity of coumarins was also reported against vertebrate parasites, such as cattle nematode *Cooperia punctata* [16]. Recently, we have proved the antifungal activities of coumarin derivatives in vitro against four fungal plant pathogens: *Macrophomina phaseolina*, *Sclerotinia sclerotiorum*, *Fusarium oxysporum* f. sp. *lycopersici*, and *F. culmorum*. The most active coumarin derivatives were shown as environmentally friendly since they were not harmful to soil-beneficial nematodes and bacteria [17]. In addition, Schiff bases are shown to be active antifungal agents [18,19,20]. Fungal plant pathogens are divided into two main groups: biotrophic pathogens, which can persist in living tissues, and necrotrophic pathogens, which kill the tissue to extract nutrients [1]. Therefore, there are two types of fungicides: eradicant fungicides that inhibit the growth of fungi organisms in already invaded plants, and protectant fungicides, which hold back fungal growth prior to the infection and prevent the organisms from entering into plants [4]. Possible targets for research of the mechanism of action of antifungal agents are enzymes responsible for the fungal growth such as demethylase [21] chitinase [22], transferase [23], or fungal exocellular enzymes; including cellulose, hemi-cellulose, pectin, lignin, and cutin; that are capable of degrading plant cell wall components, the plant cell wall, and the cuticle [24].

Microorganisms play a major role in natural soil processes that contribute to maintaining soil health and fertility. The application of pesticides has a detrimental effect on nontarget organisms and microorganisms such as bacteria. Entomopathogenic nematodes (EPNs) are beneficial soil invertebrates used in insect pest management programs [25]. EPNs serve as an excellent model to investigate the selectivity of pesticides in addition to their comparative value to other often-used methods [26]. The action of some active components of plant protection products is related to inhibition of the enzyme acetylcholinesterase (AChE), which catalyzes acetylcholine hydrolysis, an essential neurotransmitter in the central nervous system of insects, rodents, and humans [27,28]. The differences in survival percentage of EPNs in the families *Steinernematidae* and *Heterorhabditidae* may be attributed to the differences in nematode’s AChE concentration [29].

The invention of new active ingredients in plant protection products must be highly specific and environmentally and toxicologically acceptable. Crop protection by new chemicals demands extensive environmental impact assessment. Recently, we have synthesized novel coumarinyl Schiff bases using deep eutectic solvents (DESs) as environmentally friendly media. DESs are low-toxic, biodegradable solvents, which can be easily recycled and reused [30]. The aim of this study was to evaluate the inhibitory effects of these compounds against plant pathogenic fungi.

However, the application of these compounds as active substances of plant protection products and their residues could threaten the environment and human health. To design efficient and environmentally friendly antifungal agents, their environmental impact was assessed against the soil-beneficial bacteria and nematodes. Their toxicity was estimated using a computer program based on QSAR methodology. Three enzymes responsible for the fungal growth and the three plant cell-wall-degrading enzymes were chosen as targets for molecular docking. To suggest the possible inhibitory mechanism of the tested compounds against plant pathogenic fungi, experimentally obtained results were compared with the results of molecular docking on these six enzymes. In vitro enzyme inhibition study should confirm the relationship of nematicidal activity with the inhibitory capability of AChE. Molecular docking was used to determine the binding affinity and interactions of the most active compounds with AChE and reveal their mode of action.

## 2. Results and Discussion

The structures of the analyzed compounds are presented in Table 1. Details of synthesis and characterization of compounds were described previously [30].

The results of antifungal, antibacterial, and nematicidal activity evaluation of nine coumarinyl Schiff bases are presented in Table 2.

Tested compounds were most active against *Macrophomina phaseolina*. The inhibition rates 48 h after inoculation varied from 65.17% (**8**) to 71.51% (**3**) compared to control (Table 2). The most active compounds against *M. phaseolina*, **3** and **4**, are pyrazole-based Schiff bases with the aromatic rings as substituents. A study of Guo et al. [19] also showed that Schiff bases with an aromatic nucleus have enhanced antifungal activity. The substitution of bromine (compound **3**, 71.51% inhibition) or methoxy group (compound **4**, 70.36% inhibition) from position R_1_ (Table 1) with the stronger electron-withdrawing group, chlorine atom (compound **2**), lowered the inhibition against *M. phaseolina* to 67.47%, as well as two strong electron-withdrawing groups at the position R_1_ and R_2_ (compound **1**, 67.47%). The most effective compounds against *Sclerotinia sclerotiorum* were **9** (56.69%) and **4** (55.33%). Compound **9** is Schiff base with 4-methoxybenzylidene. Substitution of 4-methoxy group on benzylidene with the stronger electron-withdrawing group, dimethylamine, lowers the inhibition of compound **8** to 20.49 %. Coumarins with an electron-withdrawing group (NO_2_ or acetate) that favor the inhibition have shown an opposite effect on the human pathogen strain *Aspergilus* [31]. Fungal growth inhibitions against the *Fusarium oxysporum* f. sp. *lycopersici*, and *F. culmorum* were lower than 37.26%. None of the compounds stimulated growth of mycelia of pathogenic fungi. The highest inhibitory effect (30.34%) on *F. oxysporum* f. sp. *lycopersici* had compound **5** with chlorine and iodine atoms on two phenyl rings.

Principal component analysis (PCA) was used for easier interpretation of the inhibitory effects of compounds on individual fungi. The results of PCA analysis are presented in Table 3 and Table 4, and visualized by biplot (Figure 1). A biplot represents the scores of the observations (compounds) on the principal components, and it uses vectors (fungi species) to represent the coefficients of the variables on the principal components [32]. PCA analysis shows that the first two components explain 87.20% of the total standard variation. PC 1 is associated with *M. phaseolina*, *F. oxysporum* f. sp. *Lycopersici,* and *F. culmorum*, while PC 2 is linked to *S. sclerotiorum*. The biplot of the first two principal components (PCs) reveals that compounds **2**, **4**, **5**, and **7**, which are positioned on the top right part of the PCA biplot, influence the two fungi *F. oxysporum* f. sp. *lycopersici* and *M. phaseolina. F. culmorum* is situated at the right bottom part and is correlated with compounds **1** and **3**. *S. sclerotiorum* is an independent vector forming a large angle regarding a vector of *F. culmorum*, and angles close to 90° considering vectors of *M. phaseolina*, *Fusarium oxysporum* f. sp. *lycopersici*. Compounds that are arranged on the left top part of biplot (**9** and **6**) were highly influenced on inhibition of *S. sclerotiorum*. Compound **8**, which is located alone on the left bottom quadrant, did not show significant inhibition of any fungi.

Previously studies also have proved the antifungal activity of coumarins and Schiff bases. Chitosan-derived Schiff bases inhibited *Colletotrichum lagenarium* and *Botrytis cinerea* mycelial growth by 26–33% and 35–38%, respectively [33]. Chen et al. [34] showed that the inulin derivatives modified by Schiff bases had significantly better antifungal activity against important plant pathogens *Botrytis cinerea*, *Phomopsis asparagi*, and *Fusarium oxysporum* f. sp. *cucumerinum* compared to activity of pure inulin. In addition, the antifungal activity of Schiff bases of inulin that bears a pyridine ring exhibited good antifungal activities against *Botrytis cinerea*, but the different position of the N atom on the pyridine ring did not show significant influence on the antifungal activity [20]. The QSAR study of eight substituted coumarin derivatives of the antifungal activity against phytopathogenic fungi *Valsa mali* revealed that small electron-withdrawing substituents of coumarin’s phenyl ring and hydrophilic electron-donating groups on the coumarin’s pyrone ring could enhance the antifungal activity [8].

To elucidate the possible mechanism of action of coumarinyl Schiff bases against pathogenic fungi, a molecular docking study was performed on two groups of enzymes: enzymes responsible for the fungal growth (demethylase [21], chitinase [35], transferase [36]) and plant cell-wall-degrading enzymes (endoglucanase I [37]; proteinase K [38]; pectinase [39]). The compounds were ranked by an energy-based scoring function. The results of molecular docking on enzymes related to the antifungal activity are presented in Table 5.

A comparison of the results shown in Table 5 with the antifungal activity against *M. phaseolina* (Table 2), reveals that the best docking scores were obtained for the wall-degrading enzymes endoglucanase I. In addition, interactions of the best scored compound **3** with amino acid residuals of endoglucanase I generated the lowest total binding energy of all docked enzymes (−160.23 kcal/mol). Compound **3** exhibited the highest inhibition against *M. phaseolina*. In addition, compounds **7** and **4**, which are among the first three compounds with the highest inhibition activity against *M. phaseolina*, have binding energies which are among the first four binding energies on endoglucanase I (−146.66 and −145.73 kcal/mol, respectively). Compound **3** also released the highest binding energy by forming complexes with the pectinase. The same results were obtained in our previous study of antifungal activity of coumarin derivatives, where the compounds that inhibited *M. phaseolina* most strongly had high binding energies on the plant cell-wall-degrading enzymes, endoglucanase and pectinase [17]. Endoglucanase I is an enzyme involved in the degradation of cellulose by fungal species that catalyzes the hydrolysis of the α-1,4-glycosidic linkages of cellulose [37]. An ultrastructural microscopic observation showed that host cell penetration by *M. phaseolina* is very precise [40]. *M. phaseolina* excretes endoglucanase, which is rapidly transported through the xylem of an infected plant [41]. Endoglucanase activity from *M. phaseolina* was detected after 11 days of cultivation [24]. The amino acid sequence revealed a unique pathogen-specific endoglucanase (egl1) gene in addition to the endoglucanase gene (egl2) commonly found in *M. phaseolina* [42]. Endonogluconasa is also present in *Fusarium oxysporum*, especially in strain H57-1 [43]. Endo-1,4-glucanase (Cel12A), which belongs to the glycoside hydrolase family 12, was isolated from *Trichoderma reesei* Cel12A showed a strong hydrolysis against xyloglucan and (1 → 3,1 → 4)-*β*-glucan, the major polysaccharides of the cell wall [44]. It is important to note that a large amount of other hydrolytic enzymes for degrading cell wall polysaccharides, such as exocellobiohydrolases, β-glucosidases, as well as enzymes for lignin degradation (laccases, lignin peroxidases, galactose oxidases, chloroperoxidases, haloperoxidases, and heme peroxidases), which allows penetrating into the host tissue [45], was detected in *M. phaseolina*.

The docking study elucidated the mode of binding of the best inhibitor of *M. phaseolina*, compound **3**, with the endoglucanase I. Binding site was defined according to the crystal structure of the complex with 4-(β-D-glucopyranosyloxy)-2,2-dihydroxybutyl propanoate (PDB ID: IN1). Energies of the main interactions between the binding site residues and ligand **3** in docked pose two are tabulated in Table 6.

Figure 2a illustrates the position of the docked compound **3** into the binding site of endoglucanase I presented in the form of a hydrophobic surface. Figure 2b shows a 2D diagram of the main interaction with amino acid residues. The substrate-binding groove is located deeply at the bottom of the canyon, which is approximately 50 Å long and formed by the β-strands. The nucleophile, GLU-197, hydrogen bonds to an adjacent aspartate residue, ASP-199. Prior to a catalytic cycle, ASP-199 is in the protonated form, while GLU-197 is negatively charged [36]. Compound **3** is docked into the hydrophilic groove forming a conventional hydrogen bond TYR-171 through a hydrazide nitrogen atom (2.82 Å). One of the phenol rings creates strong π–σ interaction with TYR-177 (3.94 Å). Compound **3** forms the strongest van der Waals interaction with ASP-199, TYR-171, ASP-173, TYR-145, and ASP-199. Aromatic rings attached to the pyrazole ring, as well as benzopyran, generate several π-cation, π–σ interactions, π-π interactions, and π-alkyl interactions with surrounding amino acid residues, supporting the findings of the antifungal activity assay and literary evidence about the importance of aromatic rings in the structure for the enhanced activity [19].

Tested coumarin derivatives did not inhibit the growth of members of the beneficial bacterial soil population, *Bacillus mycoides* and *Bradyrhizobium japonicum* (Table 2). This result is contrary to similar research by Dekić et al. [12] and Chen et al. [10], where the antibacterial efficacy of coumarin and coumarin derivatives with all tested bacteria was determined. This difference in antibacterial activity can be attributed to different coumarin structures.

Nematicidal activities of tested coumarin derivatives (expressed as % of inhibition) are presented in Table 2. Most of the compounds did not exhibit nematicidal activity against infective juveniles (IJs) of two beneficial nematode species, *H. bacteriophora* and *S. feltiae*. However, compound **9** was lethal for 46.25% IJs *H. bacteriophora* and 40.00% IJs *S. feltiae* after 48 h of incubation. Compound **2** was lethal for 8.75% IJs *H. bacteriophora* and 20.00% IJs *S. feltiae*. Compound **6** caused mortality for only 2.5% IJs *S. feltiae*. Previously, we identified several coumarin derivates thatcould potentially serve as new effective nematicides [17]. Specific phenolic compounds and coumarins have been reported by other authors as nematotoxic or affecting the development stages of the entomopathogenic nematodes [46,47,48,49,50]. The most active compounds potentially pose a threat to beneficial organisms in the rhizosphere, since entomopathogenic nematodes are considered bioinsecticides. Further bioassays should include other nematode-trophic groups to identify selectivity of the most active coumarins. To better understand the mechanism of the nematicidal effect of analyzed compounds, AchE inhibition assay was performed. The AChE inhibition assay showed a low potency of the tested compounds as AChE inhibitors compared to standardized inhibitor donepezil (Table 2). However, compound **9** was an exception with 31.45% inhibition of the enzyme. Certain derivatives of Schiff bases showed pronounced activity as AChE inhibitors [51] as well as butyrylcholinesterase (BuChE) and monoamine oxidase (MAO) [52,53]. Obtained results are in correlation with the nematicidal activity assays where the same compound displayed the highest activity. As expected, obtained results confirmed a relation between inhibition of AChE and nematicidal activity, since this enzyme plays a critical role in terminating nerve impulses by hydrolyzing the neurotransmitter, acetylcholine (ACh) in the cholinergic nervous system of nematodes [54].

To understand the binding modes of tested coumarinyl Schiff bases and interactions with AChE at a molecular level, we performed blind-docking studies using AutoDock Vina software. The analysis of the lowest-energy poses of docked compounds showed their affinity to bind to the enzyme at two different sites. Compounds **9** and **8** bound to the active site with the binding energy of −7.8 kcal/mol and −7.6 kcal/mol, respectively, while compounds **7**, **3**, **1**, **2**, **6**, **4**, and **5** bound to the sites located away from (distal to) the enzyme active site with the binding energies of −9.6 kcal/mol, −9.4 kcal/mol, −9.1 kcal/mol, −9.1 kcal/mol, −9.0 kcal/mol, −8.9 kcal/mol, and −8.5 kcal/mol, respectively. Compound **9**, with the lowest-energy pose at the enzyme active site, was used for further understanding of the binding mode with AChE. The binding orientation of compound **9** showed that it is placed near the bottom of the active site gorge (Figure 3a), interacting with amino acid residues of the anionic site (AS), acyl pocket (AP), and peripheral anionic site (PAS) of AChE. The AS is responsible for positioning acetylcholine in the enzyme cavity and as a site for possible inhibitor binding. The AP guarantees the specificity of the enzyme to the substrate by preventing the entry of larger molecules into the catalytic site, while PAS portion represents a region that is important for the binding of many inhibitors [55]. According to Figure 3b, PAS residues TYR-70, ASP-72, and TYR-334 formed a van der Waals interaction, an attractive charge interaction, and a π–π T-shaped interaction with compound **9**, respectively. Amino acid residues PHE-330 and PHE-331 from the AChE AS interacted with compound **9** via van der Waals and carbon hydrogen interactions. Only one residue, PHE-288 from the AP, formed a van der Waals interaction with the phenyl ring of compound **9**. Furthermore, interactions with compound **9** included hydrogen bonds with VAL-71, GLU-73, GLN-74, and GLN-272 which probably contributed to binding site stabilization. Additionally, van der Waals interactions were noted between compound **9** and residues ILE-275, ASP-276 and ILE-287. The involvement of the mentioned residues from the AS, AP, and PAS was confirmed to contribute stability to the complex between AChE and synthetic inhibitors such as coumarin-triazole-amino acid hybrids [55], tyrosol 1,2,3-triazole analogs [56], coumarin-3-carboxamide-N-morpholine hybrids [57], chromone derivatives [28], and chromenyl coumarate [58].

According to the rules for pesticide-likeness defined by Hao et al. [59], six molecular descriptors describe the distribution of pesticides. According to that rule, a pesticide-like compound should have: MW ≤ 435 Da; LOGP ≤ 6; HBA ≤ 6; HBD ≤ 2; RB ≤ 9; AB ≤ 17. Calculated descriptors for the nine coumarinyl Schiff bases are presented in the Table 7.

None of the compounds completely satisfied pesticide-likeness criteria. Only compounds **8** and **9** had satisfactory molecular weight, but their number of hydrogen-bond acceptors was higher for one than the recommended six. The number of aromatic bonds of these compounds was also acceptable. Other compounds have high molecular weight and number of aromatics bond. The pesticide-likeness study revealed that a decrease of MW is associated with a toxic reduction of the pesticide. Aromatic double bonds in the structure are usually associated with the photostability of pesticides, one of the most important properties of pesticide molecules [59,60]. From 1970 until the year 2000, the toxicity of pesticides was reduced by introducing the leading compounds with increased MW, which indirectly induced an increase in values for the other constitutive properties.

The toxicity of compounds estimated by program T.E.S.T. is presented in Table 8. According to the acute systemic toxicity classification based on oral LD_50_ values for rats and recommended by the Organization for Economic Co-operation and Development (OECD), all compounds have estimated LD_50_ values into the range of 300–2000 mg/kg, and characterized as “harmful if swallowed”, except compound **5**, which is the least harmful (LD_50_ = 2206.90 mg/kg) and characterized as “may be harmful if swallowed” [61]. Only two, the most lipophilic compounds, **5** and **7** (Table 7), were evaluated as no mutagenic. Highest aquatic toxicity evaluated against the ciliate model organism *Tetrahymena pyriformis* was estimated for the least lipophilic compound **9** (Table 7). Compound **9** also proved to be the only lethal compound for nematode species *H. bacteriophora* and *S. feltiae*. Highly molecular weight and lipophilic compounds **1**–**5**, and **7**, proved to be highly toxic for the fathead minnow. Bioaccumulation is a process of absorption of compounds in an organism from the natural environment. All compounds had a logBAF of less than three, and their bioaccumulation was not significant [62].

## 3. Materials and Methods

### 3.1. Synthesis of Coumarinyl Schiff Bases

The synthesis of coumarin derivatives was performed in environmentally safe organic solvents [30].

### 3.2. Biological Assays

#### 3.2.1. Antifungal Assays

For the preparation of stock solutions of compound concentration 4 μmol/mL, a corresponding mass was dissolved in 2.5 mL of DMSO and 2.5 mL of distilled water. The volume of 1 mL of stock solution was added to the mixture to produce the final compound concentration (0.08 μmol/mL) and keep the amount of DMSO in the mixture at 1%. As a control, untreated potato dextrose agar (PDA) was used. The antifungal assay was performed on four cultures of phytopathogenic fungi from the fungal collection of the Faculty of Agrobiotehnical Sciences Osijek, (*Fusarium oxysporum* f. sp. *lycopersici*, *Fusarium culmorum*, *Macrophomina phaseolina,* and *Sclerotinia sclerotiourum*). The test was carried out according to the method of Siber et al. [63]. Petri dishes were kept in a growth chamber at 22 ± 1 °C, with a 12 h light/12 h dark regime. Each measurement consisted of four replicates. The radial growth of the fungal colonies was measured 48 h after inoculation. The in vitro inhibiting effects of the test compounds on the fungi were calculated by the antifungal index (% inhibition) [64].

#### 3.2.2. Antibacterial Assays

Broth microdilution method was applied to test antibacterial activity and determine the minimum inhibitory concentration (MIC) of *Bacillus mycoides* (Gram-negative) and *Bradyrhizobium japonicum* (Gram-positive) (collection of the Faculty of Agrobiotehnical Sciences Osijek), which are included in the assay. Pure bacterial cultures were obtained by inoculation on nutrient (Liofilchem, Roseto degli Abruzzi, Italy) and Vincent agar. The stock solution of coumarinyl Schiff bases was prepared by dissolving 1.024 mg in 20 μL of DMSO with the addition of sterile water up to 200 μL. The solution was diluted from 512 to 1 μg/mL with sterile broth in 96-well plate. Inoculation was performed by adding 50 μL of pure bacterial cells with a density of 1.5 × 10^8^ colony forming units (CFU/mL). Results were recorded after 48 h of incubation. The assay was set up in four repetitions.

#### 3.2.3. Nematicidal Assays

For the preparation of 500 μg/mL stock solutions, 2 mg of each compound was dissolved in 20 μL of DMSO and 3980 μL of distilled water containing 0.1% Triton. Inhibition of nematode motility and mortality was tested for all compounds in maximum concentration 500 μg/ mL with four repetitions in a 24-well plate.

An aliquot of approximately 100 IJs entomopathogenic nematodes, *Heterorhabditis bacteriophora* same as *Steinernema feltiae* (collection of the Faculty of Agrobiotehnical Sciences Osijek) were placed in each well containing 250 μL of the working solution. Distilled water containing DMSO and Triton was used as a control. Well plates were incubated in the dark at 24 °C. The number of motile, dead, and live nematodes was recorded after 48 h of incubation. Nematodes were observed under a microscope (40×) and considered dead when they failed to respond to physical stimuli with a probe. The values were determined as percentage corrected mortality according to the Schneider–Orelli formula.

#### 3.2.4. Acetylcholinesterase Inhibition Assays

AChE (EC 3.1.1.7, type VI-S from *Electrophorus electricus* (electric eel)), donepezil, and bovine serum albumin (BSA) were purchased from Sigma–Aldrich (St. Louis, MO, USA). Dimethyl sulfoxide (DMSO) was purchased from Merck (Darmstadt, Germany). Acetylthiocholine iodide (ATChI) was purchased from Alfa Aesar (Haverhill, MA, USA). 5,5′-Dithiobis(2-nitrobenzoic acid) (DTNB) was purchased from VWR International (Radnor, PA, USA). All other chemicals used were of analytical grade.

Acetylcholinesterase inhibition assay was conducted using the Ellman protocol [65]. Tested compounds were dissolved in DMSO and diluted with a phosphate buffer solution (pH 8.0), yielding the final concentration of DMSO in the assay less than 1%. The tested concentration of compounds in the final reaction mixture was 0.1 mM. Donepezil was used as a standard inhibitor. Enzyme stock solution was prepared by dissolving AChE in a phosphate buffer solution (pH 8.0) containing 0.1% (*w*/*v*) BSA. Reaction mixture was comprised of 10 µL of test compound solution, 130 µL of phosphate buffer, 20 µL of AChE solution (0.25 U/mL), 20 µL of DTNB solution, and 20 µL of ATChI solution. Prior to the addition of the substrate (ATChI), the mixture was preincubated for 15 min at 25 °C. The enzymatic activity measurements were conducted using a microplate reader (iMark, Bio-Rad, Hercules, CA, USA) in a span of 10 min at a wavelength of 412 nm. Inhibition activity was calculated using enzyme inhibition absorbance response using the formula:(1)$$\mathrm{Inhibition}\;\left(\%\right)=\frac{A_{0}-A_{1}}{A_{0}}\times 100$$
where A_1_ is the absorbance of the tested compound, and A_0_ is the absorbance of positive control.

### 3.3. Computational Methods

#### 3.3.1. Calculation of Pesticide-Likeness Molecular Descriptors

Molecular weight (MW), lipophilicity (MLOGP), number of hydrogen-bond donors (HBD), number of hydrogen-bond acceptors (HBA), number of rotatable bonds (RB), as well as number of aromatic bounds (AB) of tested compounds were calculated using ADMEWORKS ModelBuilder (Version 7.9.1.02011 Fujitsu Kyushu Systems Limited, Krakow, Poland).

#### 3.3.2. Calculation of Toxicity

The toxicity of compounds was calculated entering their SMILES notation into the program T.E.S.T. v.4.1 using nearest neighbor method. The toxicity was estimated by taking an average of the three chemicals in the training set that were most similar to the test chemical [66]. A lethal dose for rats (oral rat LD_50_) was expressed as the mg of the chemical per bodyweight of the rat (mg/kg bw). Aquatic toxicity of the compound was presented as the concentration of the test chemical in water in mol/L that caused 50% growth inhibition to *Tetrahymena pyriformis* after 48 h (48 h *T. pyriformis* IGC_50_) [67] as well as by concentration in water, which killed half of the fathead minnows (*Pimephales promelas*) in 96 h (European Chemicals Agency, ECHA-11-R-004.2-EN, 2011) [68]. The Ames test estimates mutagenicity of a compound that induces revertant colony growth of *Salmonella typhimurium*. Its results represent the alert for the potential carcinogenicity and/or teratogenicity [69]. Bioaccumulation factor (BAF), is the ratio of the concentration of a chemical in the tissue of an aquatic organism (fish) to its concentration in water (in liters per kilogram of tissue), expressed as logarithmic value [70].

#### 3.3.3. Molecular Docking

To determine the possible mechanism of action of derivatives against pathogenic fungi, a molecular docking study was performed on three enzymes responsible for the fungal growth: demethylase (sterol 14α-demethylase (CYP51), PDB ID: 5EAH) [21]; chitinase (PDB ID: 4TXE) [35]; transferase (N-myristoyltransferase, PDB ID: 2P6G) [36]; and the three plant cell-wall-degrading enzymes: endoglucanase I (PDB ID: 2OVW) [37]; proteinase K (PDB ID: 2PWB) [38]; and pectinase (endopolygalacturonase, PDB ID: 1CZF) [39]. Crystal coordinates were provided from Protein Data Bank (PDB, https://www.rcsb.org/, accessed on 1 November 2021). Prior to molecular docking, the 3D structures of ligands were optimized by Spartan ’08 (Wavefunction, Inc., Irvine, CA, USA, 2009), using the molecular mechanics force field (MM+) [71] and subsequently by the semiempirical AM1 method [72]. The molecular structures were optimized using the Polak–Ribiere algorithm until the root mean square gradient (RMS) was 0.001 kcal/mol per Å.

The molecular docking of compounds was performed using iGEMDOCK (BioXGEM, Taiwan). Applying the generic evolutionary method, each compound was docked into the binding site of a cocrystalized standard inhibitor using the following parameters: population size was 200, generations were 70, and the number of poses was 3. Program iGEMDOCK generates protein-compound interaction profiles based on electrostatic (E), hydrogen-bonding (H), and van der Waals (V) interactions. Compounds were ranked by combining the pharmacological interactions and energy-based scoring function: Total Energy (kcal/mol) = vdW + Hbond + Elec. Blind docking was carried out to investigate the modes of AChE inhibition for tested compounds using AutoDock Vina 1.1.2 software [73]. For this purpose, the crystal structure of AChE, PDB ID: 2C5G [74] was extracted from the PDB, while MGL Tools 1.5.6 [75] was employed to prepare structures for molecular docking. The blind docking site for the ligands on AChE was defined by a grid box with the dimensions 81 × 82 × 84 Å3, and center set at x = 4.98, y = 64.57 and z = 55.94. Docking simulation was performed with the standard 0.375Å resolution and 20 conformations were generated. Receptor-ligand interactions were visualized with BIOVIA Discovery Studio Visualizer 4.5 (Dassault Systèmes, San Diego, CA, USA).

#### 3.3.4. Statistical Analysis

Principal component analysis was performed by Statistica 14 (TIBCO Software Inc. 2020, Palo Alto, CA, USA).

## 4. Conclusions

Coumarinyl Schiff bases proved to be promising candidates in the inhibition of *Macrophomina phaseolina*. Pyrazole-based Schiff bases with the two aromatic rings with the weaker electron-withdrawing groups enhanced antifungal activity. The most effective antifungal compounds, **3** and **4**, did not show negative effects against beneficial soil bacteria and nematodes. Their moderate lipophilicity and small molar weight do not pose a hazard for bioaccumulation in animals and humans. The possible inhibitory mechanism of the tested compounds against *M. phaseolina* is related to the inhibition of enzyme endoglucanase, which participates in hydrolysis of cellulose. The most active compound **3**, released the highest energy binding in the substrate-binding groove of endonuclease. In addition, the relation between nematicidal activity and inhibition of acetylcholinesterase was demonstrated. Compound nine was identified as nematotoxic and an AChE inhibitor in nematodes. This compound, as the lowest lipophilic, had the highest toxicity for the aquatic model organism. However, further toxicological studies are needed to support the development of new effective nematicides or anthelminthic drugs safe for mammals. Future bioassays should involve diverse species of target organisms to identify selectivity of the most active compounds.

Based on the knowledge gained by this study, the further development of coumarinyl Schiff bases as future plant protection products will continue. The future compounds should be more effective against pathogenic fungus and other pathogenic microorganisms without negative effects on beneficial organisms, environment pollution and human health.

## Figures and Tables

**Figure 1 molecules-27-02196-f001:**
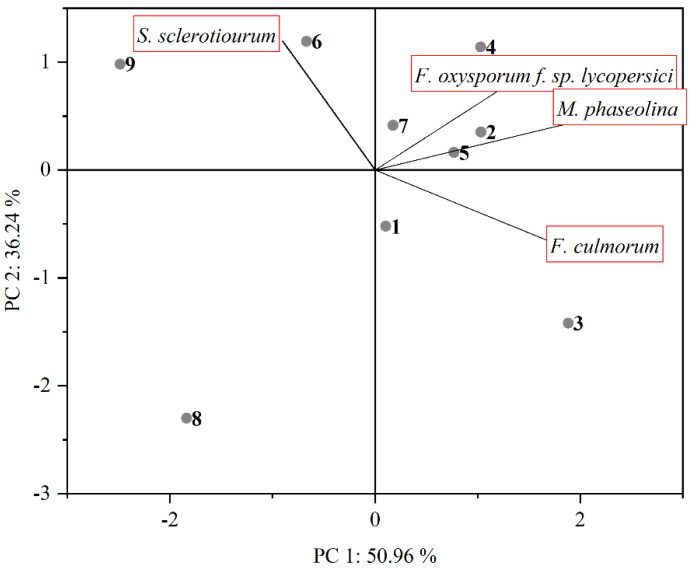
Biplot graph with first two principal components for the plant pathogenic fungi and compounds.

**Figure 2 molecules-27-02196-f002:**
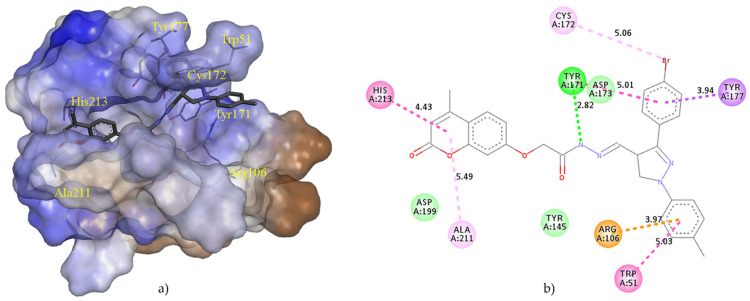
The main interactions of compound **3** with residues in the binding site of endoglucanase I (pdb ID: 2OVW): (**a**) Hydrophobic surface representation of endoglucanase I active site with docked compound **3**. Hydrophobicity in range: 3 (brown) to −3 (blue); (**b**) Two-dimensional diagram of main interactions with interatomic distances (Å). (green = conventional hydrogen bond; light green = van der Waals; brown = π-cation; dark purple = π–σ interactions; light purple = π-π interactions; pink = alkyl and π-alkyl interactions).

**Figure 3 molecules-27-02196-f003:**
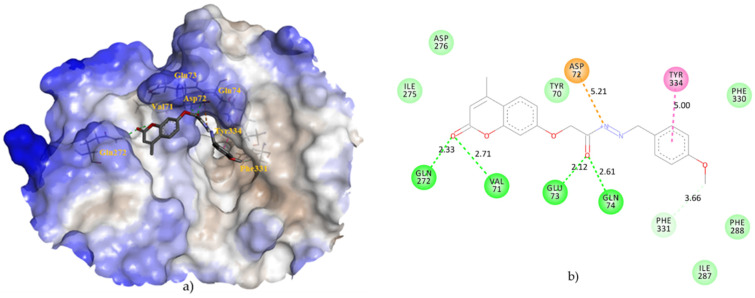
The main interactions of compound **9** with residues in the binding site of AChE (pdb ID: 2C5G): (**a**) Hydrophobic surface representation of AChE active site with docked compound **9**. Hydrophobicity in range: 3 (brown) to −3 (blue); (**b**) Two-dimensional diagram of main interactions with interatomic distances (Å). Green = conventional hydrogen bonds; light green = van der Waals; pale green = carbon hydrogen bond; brown = attractive charge; pink = π–π T-shaped interaction.

**Table 1 molecules-27-02196-t001:** Structures of the analyzed coumarinyl Schiff bases.

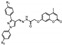	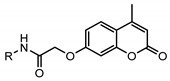
**Numb.**	**R_1_/R_2_**	**Numb.**	**R**
**1**	R_1_ = NO_2_; R_2_ = Cl	**6**	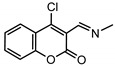
**2**	R_1_ = Cl; R_2_ = CH_3_	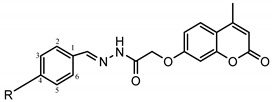
**3**	R_1_ = Br; R_2_ = CH_3_	**Numb.**	**R**
**4**	R_1_ = OCH_3_; R_2_ = CH_3_	**8**	−N(CH_3_)_2_
**5**	R_1_ = I; R_2_ = Cl	**9**	−OCH_3_
**7**	R_1_ = I; R_2_ = CH_3_	

**Table 2 molecules-27-02196-t002:** Results of biological activity assays of nine coumarinyl Schiff bases: ^a^ inhibition rate %, 48 h after inoculation at the concentration 0.08 μmol/mL; ^b^ minimum inhibitory concentration (MIC/μg mL^−1^; ^c^ percentage corrected mortality, %, 48 h after inoculation at the concentration 500 μg/mL), and acetylcholinesterase (AChE) inhibition assay.

	Antifungal Activity ^a^				Antibacterial Activity ^b^		Nematicidal Activity ^c^		Inhibition of AChE/% **
Mol. No.	*Macrophomina phaseolina*	*Sclerotinia sclerotiorum*	*Fusarium oxysporum* f. sp. *lycopersici*	*Fusarium culmorum*	*Bacillus mycoides*	*Bradyrhizobium japonicum*	*Heterorhabditis bacteriophora*	*Steinernema feltiae*	
**1**	67.47 ± 0.49	26.64 ± 3.44	20.63 ± 4.60	27.73 ± 8.20	>512	>512	0.00 ± 0.00	0.00 ± 0.00	0.00
**2**	67.47 ± 0.37	29.37 ± 6.06	24.27 ± 1.58	22.53 ± 9.12	>512	>512	8.75 ± 2.50	20.00 ± 0.00	0.00
**3**	71.51 ± 0.37	9.56 ± 5.23	23.06 ± 2.43	37.26 ± 6.01	>512	>512	0.00 ± 0.00	0.00 ± 0.00	0.00
**4**	70.36 ± 0.64	55.33 ± 9.56	26.70 ± 2.81	28.60 ± 4.90	>512	>512	0.00 ± 0.00	0.00 ± 0.00	0.00
**5**	68.05 ± 0.56	30.74 ± 11.01	30.34 ± 2.43	34.66 ± 1.74	>512	>512	0.00 ± 0.00	0.00 ± 0.00	0.00
**6**	67.47 ± 0.52	51.23 ± 9.03	25.49 ± 6.11	9.53 ± 0.71	>512	>512	0.00 ± 0.00	2.50 ± 0.50	0.00
**7**	69.20 ± 0.19	38.93 ± 6.06	21.84 ± 2.80	19.06 ± 1.99	>512	>512	0.00 ± 0.00	0.00 ± 0.00	0.00
**8**	65.17 ± 0.19	20.49 ± 6.87	9.71 ± 2.76	22.53 ± 7.93	>512	>512	0.00 ± 0.00	0.00 ± 0.00	0.59
**9**	66.32 ± 0.27	56.69 ± 7.52	16.99 ± 4.86	3.47 ± 0.19	>512	>512	46.25 ± 11.09	40.00 ± 7.07	31.45
control	0.00	0.00	0.00	0.00	0.00	0.00	0.00	0.00	donepezil 99.89

* The results were expressed as mean ± standard deviation (SD). ** Concentration of compounds in the final reaction mixture was 0.1 mM.

**Table 3 molecules-27-02196-t003:** Explained variance using principal component analysis.

PC	Eigenvalue	% of Variance
**1**	2.04	50.96
**2**	1.45	36.24
**3**	0.32	7.98
**4**	0.19	4.81

**Table 4 molecules-27-02196-t004:** Extracted eigenvectors of first two principal components.

Fungi	Coefficients of PC1	Coefficients of PC 2
*Macrophomina phaseolina*	0.61	0.25
*Sclerotinia sclerotiorum*	−0.29	0.70
*Fusarium oxysporum* f. sp. *lycopersici*	0.49	0.54
*Fusarium culmorum*	0.55	−0.39

**Table 5 molecules-27-02196-t005:** Docking score energies (total E/kcal mol^−1^) of interactions of the best docked poses of nine coumarinyl Schiff bases in complex with: demethylase (sterol 14α-demethylase (CYP51), PDB ID: 5EAH); chitinase (PDB ID: 4TXE); transferase (N-myristoyltransferase, PDB ID: 2P6G); endoglucanase I (PDB ID: 2OVW); proteinase K (PDB ID: 2PWB); pectinase (endopolygalacturonase, PDB ID: 1CZF).

Demethylase		Chitinase		Transferase		Endoglucanase I		Proteinase K		Pectinase	
Comp. (Pose)	Total E	Comp. (Pose)	Total E	Comp. (Pose)	Total E	Comp. (Pose)	Total E	Comp. (Pose)	Total E	Comp. (Pose)	Total E
4 (0)	−131.26	2 (1)	−145.90	1 (0)	−131.17	3 (2)	−160.23	1 (1)	−148.87	3 (2)	−115.17
2 (0)	−128.04	5 (0)	−140.30	2 (1)	−111.41	1 (1)	−151.78	2 (0)	−135.07	2 (2)	−112.13
1 (0)	−124.73	7 (2)	−136.40	3 (2)	−105.96	7 (1)	−146.66	3 (1)	−124.76	8 (1)	−105.99
3 (1)	−120.73	4 (1)	−136.20	4 (1)	−118.54	4 (0)	−145.73	4 (0)	−134.79	6 (0)	−105.82
5 (1)	−118.59	1 (2)	−135.40	5 (0)	−125.71	2 (1)	−143.23	5 (0)	−135.03	5 (2)	−105.35
7 (2)	−118.57	3 (2)	−130.70	6 (1)	−103.44	5 (0)	−134.50	6 (1)	−117.53	9 (0)	−103.88
6 (0)	−113.25	8 (1)	−126.10	7 (0)	−113.52	8 (1)	−125.94	7 (0)	−120.70	7 (0)	−102.96
8 (2)	−98.68	9 (0)	−122.00	8 (1)	−106.26	6 (2)	−122.20	8 (2)	−110.73	1 (0)	−100.86
9 (1)	−97.66	6 (0)	−117.60	9 (2)	−98.78	9 (2)	−121.83	9 (2)	−113.81	4 (0)	−96.05

**Table 6 molecules-27-02196-t006:** The energies of the main interactions between compound **3** and endoglucanase I (pdb ID: 2OVW).

Residue	Energy (kcal/mol)	Residue	Energy (kcal/mol)
H Bond		π-π Interactions	
S-TRP-51	−1.00	S-TRP-51	−16.88
S-TYR-171	−4.45	S-HIS-213	−7.12
Van der Waals interactions		Alkyl and π-alkyl interactions	
S-ARG-106	−4.14	M-ALA-211	−0.27
S-TYR-145	−3.99	M-CYS-172	−5.45
S-TYR-171	−4.45	π-cation interactions	
S-ASP-173	−5.18	S-ARG-106	−10.23
S-ASP-199	−1.21	π–σ interactions	
S-HIS-209	−3.50	M-TYR-177	−7.26
S-HIS-213	−5.26	S-TYR-177	−11.84

M = main chain; S = side chain.

**Table 7 molecules-27-02196-t007:** Molecular properties of coumarinyl Schiff bases.

Compound	MW	MLOGP	HBA	HBD	RB	AB
**1**	559	4.38	11	1	5	18
**2**	528	4.70	8	1	5	18
**3**	573	4.81	8	1	5	18
**4**	524	3.93	9	1	5	18
**5**	640	4.92	8	1	5	18
**6**	438	3.61	8	1	4	12
**7**	620	4.92	8	1	5	18
**8**	379	3.19	7	1	3	12
**9**	366	2.94	7	1	3	12

MW (molecular weight): MLOGP (Moriguchi octanol–water partition coefficient); HBA (number of hydrogen-bond acceptors; HBD (number of hydrogen-bond donors); RB (number of rotatable bonds); AB (number of aromatics bonds).

**Table 8 molecules-27-02196-t008:** Estimated toxicity for nine coumarinyl Schiff bases.

Compound	Oral rat LD_50_ (mg/kg bw) ^a^	*Tetrahymena pyriformis* IGC_50_ 48-h (mg/L) ^b^	Fathead Minnow LC_50_ 96-h (mg/L) ^c^	Mutagenicity Value (Result) ^d^	Bioaccumulation (logBAF/L kg^−1^) ^e^
**1**	1050.80	7.67	8.63 × 10^−4^	0.74 (pos)	1.04
**2**	1214.11	7.25	8.15 × 10^−4^	0.75 (pos)	1.04
**3**	344.53	7.86	8.83 × 10^−4^	0.71 (pos)	1.07
**4**	1183.17	7.19	8.08 × 10^−4^	0.74 (pos)	0.66
**5**	2206.90	8.78	9.87 × 10^−4^	0.48 (neg)	1.07
**6**	847.98	14.53	0.50	0.55 (pos)	2.35
**7**	1164.29	8.50	9.56 × 10^−4^	0.43 (neg)	1.07
**8**	643.35	5.20	0.22	0.70 (pos)	1.27
**9**	623.67	5.02	0.30	0.53 (pos)	1.27

^a^ mg of compound per bodyweight of the rat required to kill half of a tested population (LD_50_); ^b^ concentration (mg/L) of compound in water that causes 50% growth inhibition to *Tetrahymena pyriformis* after 48 h (IGC_50_); ^c^ concentration (mg/L) of compound in water that kills half of fathead minnows (*Pimephales promelas*) in 96 h (LC_50_); ^d^ estimated mutagenicity of compound on *Salmonella typhimuriu*; ^e^ logarithmic value of ratio of the concentration of compound in the tissue of an aquatic organism to its concentration in water (in litres per kilogram of tissue).

## Data Availability

Data are contained within the article.

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
