# Peer review of "Effects of Coumarinyl Schiff Bases against Phytopathogenic Fungi, the Soil-Beneficial Bacteria and Entomopathogenic Nematodes: Deeper Insight into the Mechanism of Action"

_molecules, 2022, doi:10.3390/molecules27072196_

Round 1

Reviewer 1 Report

The manuscript molecules-1650261 "Biological Activities of New Coumarinyl Schiff Bases Related to the Plant Protection: Deeper Insight into the Mechanism of Action" by Rastija et al. describes the study of biological activity of previously synthesized coumarin based Schiff bases by in vitro methods and calculations.

Questions and comments:

1) The use of the phrase "New Coumarinyl Schiff Bases" in the title is incorrect, since the studied compounds were synthesized earlier and published by authors 5 years ago (ref. 26).

2) Why among all compounds from [26] were the compounds 1-9 from this manuscript chosen? The structures of the compounds 1-9 are very different, so it is not clear on the basis of what such a selection was made.

3) Why did the authors not evaluate the MBC of the obtained compounds?

4) The manuscript lacks a discussion of the obtained results on biological activity, as well as conclusions about the structure-activity relationship.

5) Although there is information in the manuscript about the calculated toxicity, experiments in vitro or in vivo should be also added.

6) The point of future prospective of obtained results should be added.

Author Response

We are thankful to the reviewer for a constructive review. We have accepted most of your remarks and revised the manuscript according to them. All changes are marked by red colour. These are our answers:

1) The use of the phrase "New Coumarinyl Schiff Bases" in the title is incorrect, since the studied compounds were synthesized earlier and published by authors 5 years ago (ref. 26).

The reviewers comment is correct, so we deleted the term „new“ from the title. Also, we have changed the title according to the remarks of both reviewers.

2) Why among all compounds from [26] were the compounds 1-9 from this manuscript chosen? The structures of the compounds 1-9 are very different, so it is not clear on the basis of what such a selection was made.

All studied compounds are coumarinyl Schiff bases, which means that their structures carrying imine (–C=N–) functional group and coumarine core. They differ in substitutes attached on imine group and coumarine core.

3) Why did the authors not evaluate the MBC of the obtained compounds?

The aim of the study was to determine the level of investigated antimicrobial agent that greatly inhibits bacterial growth. The minimum inhibitory concentration (MIC) was determined.

4) The manuscript lacks a discussion of the obtained results on biological activity, as well as conclusions about the structure-activity relationship.

The discussion was extended, as well as conclusions about the structure-activity relationship, according to the advice. All changes in the manuscript were marked in red colour.   

5) Although there is information in the manuscript about the calculated toxicity, experiments in vitro or in vivo should be also added.

As we mentioned in Introduction, the toxicity was estimated using a computer program, which is based on QSAR methodology.

The REACH (Registration, Evaluation and Authorization of Chemicals) guidelines (of the European Parliament) for animal safety restrict the extensive use of animals in testing. Animal tests can be avoided if the hazardous properties of a substance can be predicted using computer models, sometimes referred to as “in-silico” methods. Regulation suggests the QSAR approach to predict the intrinsic properties of chemicals by using various databases, theoretical models and software applications instead of conducting tests on animals. (ECHA-11-R-004.2-EN, 2011)

6) The point of future prospective of obtained results should be added.

The future prospective of obtained results was added in Conclusion, according your advice.

Reviewer 2 Report

Please refer to the attachments.

Author Response

We are thankful to the reviewer for a constructive review.. We have accepted most of your remarks and revised the manuscript according to them. All changes are marked by red colour. These are our answers:

PAGE 1

  1. The authors need to revise the title of the paper in a more meaningful way.

The title was changed according in more meaningful way, as you advice.

  1. The abstract is written in a way lacks logic. It should highlight the salient findings more critically. The authors report the anti-fungal and anti-nematode activity in the abstract, but do not mention anything about the bactericidal activity with the compound, I suggest inserting these results in the abstract.

The abstract was revised according your advice. Bactericidal activity was mentioned, but tested coumarin derivatives did not inhibit the growth of members of the beneficial bacterial soil population Bacillus mycoides and Bradyrhizobium japonicum.

In abstract we have mentioned that “Most of the compounds did not exhibit inhibitory effects against two beneficial nematodes and bacteria, except compound 9 which was lethal (46.25 %) for nematode Heterorhabditis bacteriophora.” Now, we have changed sentence in: Neither compound did not exhibit inhibitory effects against two beneficial bacteria (Bacillus mycoides and Bradyrhizobium japonicum) ), and two entomopathogenic nematodes...

  1. Keywords are present in the title, choose others.

Keywords were changed words that are not present in the title.

  1. Introduction need more convincing rational for this article. Again, the authors report relevant information about anti-fungal activity, but do not mention anything about anti-nematode and bactericidal activity with compound, I suggest inserting this information in the introduction.

The Introduction has been changed according your advice. Nematicidal and antibacterial activities were described additionally.

PAGE 2

  1. Introduction need more convincing rational for this article.

The Introduction has been changed according your advice.

PAGE 3

  1. Results not presented and not discussed.

Details of synthesis and characterization of compounds were described previously [29] Molnar, M.; Komar, M.; Brahmbhatt, H.; Babić, J.; Jokić, S.; Rastija, V. Deep eutectic solvents as convenient media for synthesis of novel coumarinyl Schiff bases and their QSAR studies. Molecules 2017 22, 1482..

PAGE 4

  1. I suggest a textual reorganization! topics 2.1.1; 2.1.2; and 2.1.3, refer to a table present in topic 2.1 (table 2).

A text was reorganized, and Table 2 and Table 3 was fused in one.

PAGE 5

  1. Authors should discuss the results integrally. The discussion is based on individual results. I suggest that integrating the results will give more value to the work. I suggest that you discuss by integrating all your results Activity Against Benefical Soil Bacteria, Activity Against Benefical Soil Nematodes and AChE Inhibition). You can use correlation tests (PCA or Pearson Correlation).

The text was reorganized and results were presented integrally. Principal component analysis (PCA) was used for easier interpretation of the inhibitory effects of compounds on individual fungi. The results of PCA analysis were presented in the Table 3 and visualized by biplot (Figure 1). Other biological activities were not included in PCA since did not exhibit inhibitory effects against two beneficial bacteria and two entomopathogenic nematodes.

PAGE 7

  1. Authors should discuss the results integrally. The discussion is based on individual results. I suggest that integrating the results will give more value to the work. I suggest that you discuss by integrating all your results. You can use correlation tests (PCA or Pearson Correlation).

The text was reorganized and results were presented integrally. Principal component analysis was performed on biological activities that were analyzed by the same methodology and expressed by the same units.

  1. The discussion is poorly written hence, needs rewriting. The discussion should be further strengthened by adding some more relevant papers. The literature search is insufficient, only few related research papers in the past three years are cited, add the latest research results appropriately. See the below links if you think it will benefit your discussion.

Discussion was extended and strengthens with more relevant paper. Mentioned link is not visible. 

PAGE 9

  1. The discussion is poorly written hence, needs rewriting. The discussion should be further strengthened by adding some more relevant papers. The literature search is insufficient, only few related research papers in the past three years are cited, add the latest research results appropriately. See the below links if you think it will benefit your discussion.

Discussion was extended and strengthens with more relevant paper. Mentioned link is not visible. 

According to the reviewer's comment, we rewrote the discussion and added the following references:

Bousada, G.M.; de Sousa, B.L.; Furlani, G.; Agrizzi, A.P.; Ferreira, P.G.; Leite, J.P.V.; Mendes, T.A.O.; Varejão, E.V.V.; Pilau, E.J.; Dos Santos, M.H. Tyrosol 1,2,3-triazole analogues as new acetylcholinesterase (AChE) inhibitors. Comput Biol Chem. 2020 88, 107359.

Tehrani, M.B.; Rezaei, Z.; Asadi, M.; Behnammanesh, H.; Nadri, H.; Afsharirad, F.; Moradi, A.; Larijani, B.; Mohammadi-Khanaposhtani, M.; Mahdavi, M. Design, Synthesis, and Cholinesterase Inhibition Assay of Coumarin-3-carboxamide-N-morpholine Hybrids as New Anti-Alzheimer Agents. Chem Biodivers. 2019 16, e1900144.

Baruah, P.; Basumatary, G.; Yesylevskyy, S.O.; Aguan, K.; Bez, G.; Mitra, S. Novel coumarin derivatives as potent acetylcholinesterase inhibitors: insight into efficacy, mode and site of inhibition. J Biomol Struct Dyn. 2019 37, 1750-1765.

Li, W.; Li, J.; Shen, H.; Cheng, J.; Li, Z.; Xu, X. Synthesis, nematicidal activity and docking study of novel chromone derivatives containing substituted pyrazole. Chin Chem Lett 2018 29, 911-914.

PAGE 10

  1. Provide experimental work plan at the start of M&M. No detail description is available about the experimental design. What statistical method is used?

Work plan was schematically presented in graphical abstract. Statistical methods was described in Material and methods.

PAGE 12

  1. Rewrite the conclusion! It needs to be much improved.

Conclusion was rewritten. The future prospective of obtained results was added in Conclusion, according your advice.

PAGE 13

  1. Has there been any request for and authorization from the ethics committee for animal experimentation, for toxicity assessments in rats? It is imperative to enter the authorization number.

Nematodes are invertebrates and do not need ethical clearence. In silico methods were performed to calculate toxicity in rats. As we mentioned in Introduction, the toxicity was estimated using a computer program, which is based on QSAR methodology.

Round 2

Reviewer 1 Report

I thank the authors for answering my questions and improving the manuscript. 

Reviewer 2 Report

Thanks for attending the suggestions. The manuscript has been significantly improved. In view of the above, I believe that the article presents robust and consolidated content.